# Types, method, and mode of implementation of pain/symptom maps in musculoskeletal pain rehabilitation: A scoping review protocol

**Ukponaye Desmond Eboigbe** [ID]*, **Aliyu Lawan, Alison Rushton** [ID], **David M. Walton**

School of Physical Therapy, Faculty of Health Sciences, University of Western Ontario, London, Canada

* ueboigbe@uwo.ca

## Abstract

### Introduction

Pain maps are tools used for assessing the extent, location, or distribution of pain or symptoms for clinical or research purposes. Pain mapping involves a transformational representation of patients' experiences of pain into a graphical, numerical, or descriptive form that typically requires a patient to indicate the affected body regions and may include additional information such as qualitative description or intensity. In preparation for innovative technology-enabled development of quantifiable pain maps, this review will focus on the methodological aspects of recent pain maps in addition to the reported measurement properties of each mapping approach. This will identify current gaps in knowledge, consistencies in implementation, and inform directions for future development of more person-centric and meaningful pain maps. The objective of this scoping review is to explore the commonly used types of pain/symptom maps in musculoskeletal pain by classifying design (types) across five categorical features: scalability, region-specificity, aspect or orientation, segmentation, and sex identification, and investigate their methods and modes of implementation.

### Methods

Key sources of evidence such as Medline, Embase, PsycINFO, CINAHL, Scopus, Web of Science, will be searched from inception to June 5, 2024, including grey literature from reference screening, library and organizational collections such as WorldCat, ProQuest Global Dissertation, Google Scholar, and Google to find descriptions or evaluations of pain/symptom maps in people with pain of a primarily musculoskeletal origin. Studies reporting standard patient-reported pain or body mapping interventions will be considered but studies that present X-ray or CT or MRI scans or artistic body maps will be excluded. Primary outcomes include 'types' of design: scale, segments, sex, orientation, region; pain mapping methods: marking, shading, checking; and mode of implementation: paper, digital, etc. Secondary outcomes include axis I: pain location, extent or distribution; and axis II: pain severity, intensity, and quality. Eligibility screening and data extraction will be conducted by two independent reviewers. The review is intended to initiate research that

**Data availability statement:** No datasets were generated or analysed during the current study. All relevant data from this study will be made available upon study completion.

**Funding:** The author(s) received no specific funding for this work.

**Competing interests:** The authors have declared that no competing interests exist.

promotes the integration of data-friendly solutions and supports the application of machine learning in musculoskeletal pain evaluation.

## Introduction

Pain is subjective and multidimensional, making it a complex phenomenon to measure for research purposes. Musculoskeletal pain refers to pain that is primarily localized to muscles, tendons, ligaments, joints, and bones, and may or may not include referred (secondary) pain in anatomically-predictable areas [1]. Musculoskeletal conditions are typically characterized by persistent pain that limits mobility and functionality and inhibits people's ability to work and participate in society [2]. Musculoskeletal pain is a global epidemic, and a major determinant of years lived with disability [3]. The burden from musculoskeletal pain is predicted to increase globally especially in low- and middle-income countries due to population growth, aging, and increasing risk factors for other musculoskeletal conditions, such as obesity, injury, and a sedentary lifestyle [4]. Although the impact of musculoskeletal pain is likely underestimated, existing evidence supports an urgency in addressing conditions of musculoskeletal pain holistically [4]. For the overall understanding of pain mechanisms and the evaluation of methods to control pain, accurate and purposeful measurement of pain characteristics is crucial [5]. However, beyond the ubiquitous Numeric Pain Rating Scale, there is little consistency or standardization in the way pain is measured and understood in pain research [6].

Pain (or body) maps are tools used for assessing pain/symptom characteristics including pain extent, location, or distribution, and can be used for clinical or research purposes [7]. They require a transformational representation of pain experiences into a graphical, numerical, or descriptive form that typically requires patients to indicate their region of pain and may include additional information such as qualitative or intensity aspects of the pain. The mapping 'method' refers to the specific procedures or techniques used to assess and capture the spatial characteristics of pain (e.g., shading, marking, grid, etc.), while the 'mode' of assessment refers to the mediums or platforms through which the patient interacts with the pain mapping tool (e.g., paper and pen, digital, etc.) [8]. Pain mapping has gained more recognition in recent times and the methods and modes of implementation have also evolved from pen-on-paper to computer-generated images, web-based applications, and mobile applications [9]. Different methods and modes of implementation of pain maps have been described, from relatively simple checklists or two-dimensional line drawings to interactive three-dimensional digital maps [9]. The methods and modes of implementing pain mapping, such as the resolution of the selectable area (e.g., individual pixels vs. anatomic regions) have a considerable impact on the resulting spatial estimate of pain characteristics [10]. There are many potentially important inconsistencies in how pain maps are built, administered, and interpreted [11]. For example, the Collaborative Health Outcome Information Registry (CHOIR) body map contains 36 anterior and 38 posterior coded body segments highlighting anatomical or pain conditions with sex demarcation [12], which is quite different from the Pain-QuiLT body map which has over 100 codified body locations [13]. Others such as the Corlett and Bishop body map, Michigan body map, and Nordic body map each have their unique and separate features.

Most pain maps present a body image that represents an atypical well-defined healthy-appearing, fully intact, binary male or female, with body morphometric characteristics most closely aligned with a pseudo-idealized white European body structure. We are unaware of any pain map validated and published specifically for use in people with severe obesity [14], people of Black African or Pacific Islander ancestral background, or people with amputations,

as examples. As the transformational representation required to complete a pain map requires that the respondents can 'see themselves' in the body image being presented to them, these are non-trivial considerations. The optimal methods of collecting spatial pain estimates for use in musculoskeletal pain research has yet to be adequately synthesized [15], making comparisons of different study results often complicated or impossible [16].

Machine learning (ML) and artificial intelligence (AI) are transforming pain mapping by enabling more accurate and comprehensive analysis of pain patterns beyond a simple summation of 'areas selected' or 'pixels shaded'. These technologies can process large datasets of pain maps to identify subtle trends in pain location, extent or distribution, intensity, and changes over time, helping to create personalized treatment plans. Goldstein and colleagues [17] have shown that ML/AI models have the potential to integrate patient-specific factors like demographics and medical history to reduce trial-and-error in pain management and predict the progression of musculoskeletal conditions. Furthermore, ML can contribute to the development of standardized and objective pain maps, improving consistency and accessibility for diverse populations. Sankaran and colleagues [18] argued that the integration of AI/ML with electronic health records (EHRs) promises to streamline data collection, advancing both clinical practice and research in pain management. In essence, ML and AI are driving a paradigm shift in pain mapping by offering more accurate, comprehensive, and personalized insights into pain patterns [18]. These advancements not only improve clinical outcomes but also promise to make pain management more efficient, effective, and accessible across a variety of patient populations.

As technology advances, we can expect new designs, methods, and modes of implementing pain maps. Eliav and Grace [19] proposed that for the measurement of pain experiences with pain/symptom maps to be accurate, the methods should be standardized and used consistently with each patient. We note the rapid development in electronic health management that has created a strong need for a standardized body map to efficiently collect self-reported pain data to enhance practice, research, and data integration [20]. To contribute towards standards for pain/symptom mapping it is important to understand the current state of the field, including an appreciation for the types and uses of pain maps previously published, seeking similarities and differences in their evolution. The purpose of this study is to scope the existing literature on pain maps to better understand the current state and conduct important groundwork for further work to come.

## Box 1. Definition of terminology
## Axis I (Spatial measure):

- **Pain Extent:** Pain extent refers to the area of a body chart selected when representing where the pain is experienced, typically expressed as a sum or a proportion of total region or area selected [21].

- **Pain Location:** Pain location refers to a specific point (geometric) or segment (spatial) of the body where pain is localized. It involves pinpointing the position(s) where pain is experienced, such as anterior upper arm or medial knee [15].

- **Pain Distribution:** For purposes of classifying findings from this review, pain distribution refers to the pattern or spread of pain sensation in relation to known anatomical structures of the body, which can be best described as radiating or referred pain usually bounded by known dermatomal, myotomal, sclerotomal, or peripheral nerve/artery distributions (e.g., C6 dermatome or sciatic nerve distribution). [many authors including

[15] view pain distribution as an alternative term for pain extent or a combination of pain extent and location.]

## Axis II (Nociceptive measure):

- **Pain Quality:** Pain quality refers to the qualitative or descriptive characteristics or nature of the pain sensation. It often includes descriptors such as dull/aching, burning, throbbing, stabbing, tingling, electric, numbness, cold, and itchy [22].

- **Pain intensity:** Pain intensity refers to the subjective measure that quantifies the magnitude of pain experienced. Pain intensity is most commonly measured using numeric or verbal pain rating scales, including the common 0-10 numeric pain rating scale [23].

- **Pain Severity:** For purposes of this review, pain severity will refer to a descriptive omnibus indicator of the level of discomfort or distress caused by pain, integrating concepts like intensity, bothersomeness, and mood disturbance. It can be conceptualized as the magnitude to which pain interferes with daily life routine, functionality, and overall quality of life, and is expected to be quantified through quantitative or qualitative means [24].

## Other measures:

- **Scaling:** Scaling or scalability in pain mapping refers to the ability of the pain map to change dimensions (zoom) more closely to the bodily proportions of the respondent in a digital form or have a marked dimensional scaling (legend) in paper or dummy (real manikin) model.

- **Types:** This includes identifying pain maps by characteristic features that define it, e.g., dimensions, region, segments, sex, scaling

- **Methods:** This includes processes or procedures for assessing the spatial pain characteristics, e.g., freehand-drawing or marking, grid-system, checklist, shading.

- **Modes:** This includes the medium of interaction between the patient and the pain mapping tool, e.g., paper, electronic or digital, real human doll

## Objective

The objective of this scoping review is to explore the commonly used types of pain/symptom maps in musculoskeletal pain literature by classifying across five categorical features: scalability, region-specificity, aspect or orientation, segmentation, and sex identification, and investigate the methods and modes of assessing pain characteristics in those.

## Methods

### Design

The scoping review will follow the methodological framework described by Arksey and O'Malley consisting of five stages: 1) formulating the research questions, 2) identifying relevant studies, 3) selecting eligible studies, 4) charting the data and, 5) collating, summarizing, and reporting the results [25]. To ensure rigor, transparency, and reproducibility, we produced the protocol in accordance with the preferred reporting items for systematic review and meta-analysis extension for scoping review (S1 File) guidelines [26]. This scoping review protocol was registered in the Open Science Framework (https://osf.io/vsk7h/) on May 31, 2024.

### Eligibility criteria

**Population.** Study participants experiencing pain in the musculoskeletal system. Studies that recruit participants who have pain related to infection, cancer, autoimmune disease, neurological conditions (e.g., stroke, multiple sclerosis, Amyotrophic lateral sclerosis), or other non-musculoskeletal causes will be excluded.

**Intervention.** Standard patient-reported pain or body mapping procedures, including pen-on-paper drawings, manikin (real dummy), pain questionnaires containing pain maps, and digital pain mapping with computer software or mobile apps. Studies lacking definite patient-reported pain/symptom measurement intervention such as X-ray or CT or MRI scans in pain assessment will be excluded. Additionally, studies using artistic body maps which are life-sized, mostly caricature, images that qualitatively represent life experiences with symbols at corresponding body locations whose meaning is exclusive to the creator [27], will be excluded.

**Outcomes of interest.** The primary outcomes of interest include the types, methods, and modes of implementation of the pain/symptom map. The secondary outcomes are spatial pain/symptom characteristics (Axis I: extent, location, or distribution) and nociceptive aspect (axis II: intensity, quality, or severity) assessed in the included studies. For a study to be included, it will assess any of the items in axis I.

**Study design.** All study designs will be included.

### Information sources

A systematic search in the following electronic databases will be conducted from inception to June 5, 2024: Medline, Embase, CINAHL, PsycINFO, Scopus, Web of Science, including grey literature from a database of library and institutional collections such as WorldCat, ProQuest Dissertation & Thesis Global, Google scholar and Google. Considering the broadness of the search scope, the review team anticipates that only a few articles in other languages may be relevant to our study for the following reasons: 1. the highest-impact journals in the field require at minimum an English-language translation, 2. automatic translation tools like Google translate or newer tools like ChatGPT have yet to demonstrate adequately reliable translation of high-level academic work (e.g., https://shorturl.at/djq9T) sometimes making significant errors, and 3. the nature of this review is to identify and characterize common pain maps and scoring strategies, the majority of which will have been used in more than one publication.

### Search strategy

In collaboration with a research librarian, a search strategy was developed in Medline (Ovid) which was adapted for use in the other databases. Hand-searching of citations for relevant articles will also be attempted. Using relevant keyword search, "pain" will be the overarching term and/or keyword for a variety of musculoskeletal conditions [28], and keywords for pain/symptom mapping interventions will be applied, as follows: (pain) AND ("pain map*" OR "body map*" OR "pain app*" OR "pain diagram*" OR "body diagram*" OR "pain draw*" OR "pain chart*" OR "bod* chart*" OR "symptom chart*" OR "symptom map*" OR manikin* OR mannequin*).The framework of the search strategy is reported in Appendix II (S2 File). To ensure that the search strategy was meticulously constructed, the review team held multiple consultation meetings with the Library Information Specialist that is assigned to the project.

### Selection of sources of evidence

All identified citations will be uploaded to Covidence, and the screening process will be documented in a PRISMA flowchart [26] (Fig 1), including reasons for exclusion. Duplicate citations will be removed through automatic text screening. Two independent reviewers will screen

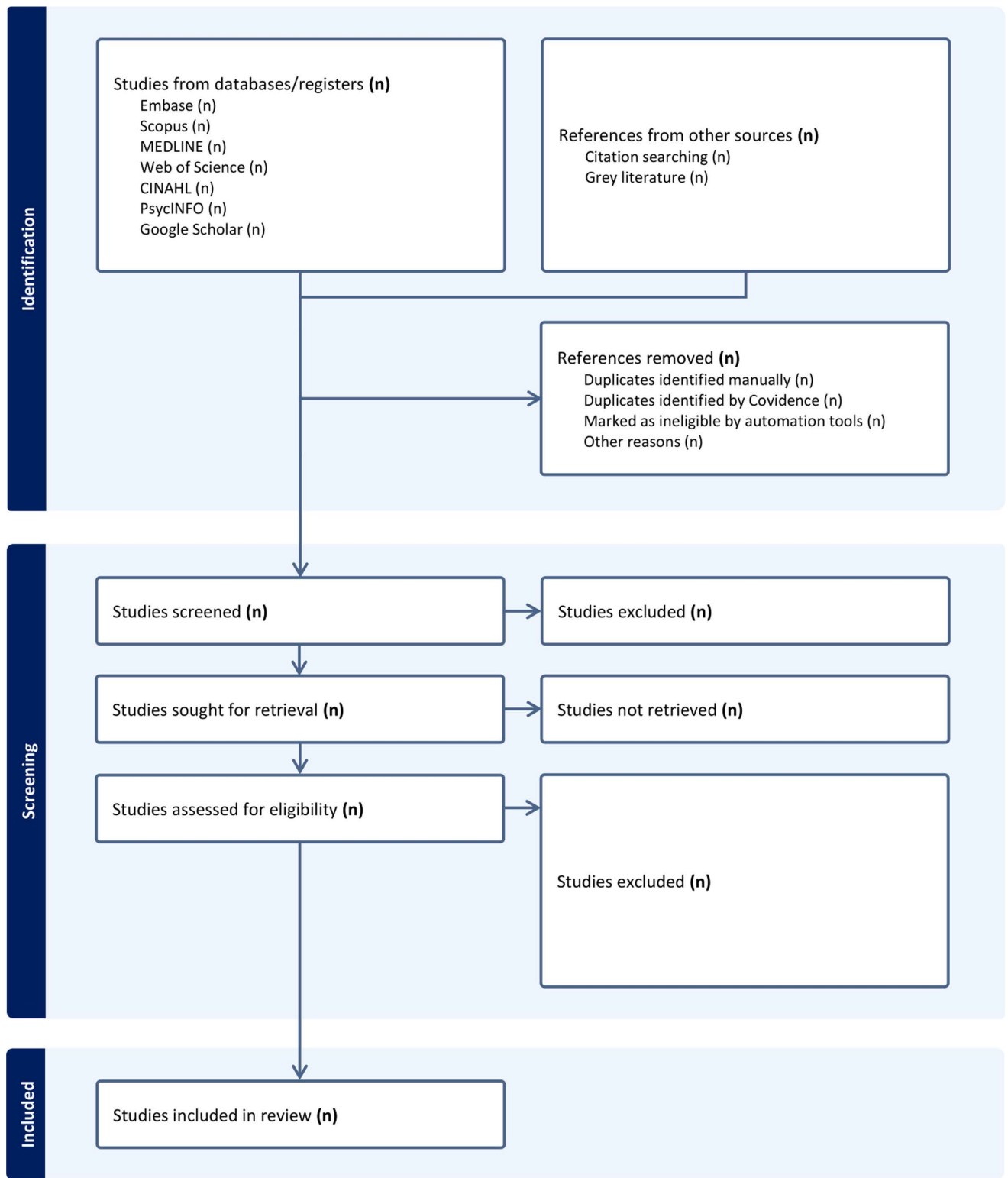

**Fig 1. Flow diagram of study selection processes.**

titles and abstracts to identify relevant studies, followed by full-text screening to assess eligibility. Disagreements will be resolved through discussion between the reviewers; if consensus is not reached, a third experienced reviewer will be consulted to make the final decision. However, the nature of the review occasioned by broadness of the topic, diversity of study designs, less focus on quality assessment, and the potential inconsistencies in framework for deploying intervention, data charting priorities, and iterative and reflective process [25], complicates defining a kappa threshold and makes it less important to report inter-rater reliability.

## Data charting

In accordance with Levac and colleagues [29], after full-text screening, the data extraction form (S3 File) will be piloted with two reviewers using 5 articles selected at random. The process involves charting key items of information obtained from the included studies by sorting material according to relevancy based on the planned analysis and evidence synthesis. The draft data extraction form may be modified and revised as necessary during the process of extracting data from the included evidence sources. Modifications might include creating new columns to assess additional details about pain mapping that were unanticipated, potentially including different methods for data analysis (description of how pain mapping data was quantified), views (lateral, dorsal, etc), scoring approach, renaming, and merging columns to adequately present findings.

## Data items

The proposed data items to be extracted for the summary data table include study author and publication dates; the purpose of the study; study design; patient characteristics: pain condition, sex, age; categorical features of pain/symptom map. The specific features to be extracted are: I. Proportional scaling features, i.e., zooming in digital designs or marked dimensional scale legend in paper or real manikin model. II. Body region the pain map fundamentally designed to assess. III. Body and/or orientational dimension of the pain mapping image, i.e., 2 or 3 in paper or digital form. IV. The number of bodily or anatomical segments in the pain map if segmented or plain if non-segmented. V. The sex attributed to the body map or generic if not indicated; methods, and modes of implementation of pain/symptom map; pain-related metrics (axis I – extent, location, or distribution; axis II - intensity, quality, severity); nomenclature for the mapping tool.

As part of data extraction, types, methods and modes of implementation of pain/symptom maps will be assessed. The types will be classified using five distinct features: scalability, region-specificity, aspect or orientation, segmentation, and sex identity as demonstrated by [12] and [30].

---

### Box 2. Characteristic features of a pain map

1. **Scalability:** features that allow for proportional adjustment of the dimensions of pain maps.

2. **Region(-specificity):** The body region the pain/symptom is designed to assess.

3. **Aspect/depth:** The n-dimensional orientation (i.e.,: 2, 3 or even more) of the pain/symptom map which allows for depth or multidirectional assessment of pain [31].

4. **Segmentation:** The number of predefined body segments in the pain/symptom map [12].

5. **Sex Identity:** The characteristic feature that allows respondents to select their sexual identity or the sexual identity(ies) that is ascribed to the pain/symptom map by design [31].

---

## Method of pain/symptom mapping:

- **Freehand or marking:** Participants draw out their pain locations freely across pixels using freehand tools [9] & [31] or marked symbols to indicate their pain [32].

- **Grid:** Participants indicate pain locations by selecting squares or boxes with predefined coordinates that define the body map [33] or selecting entire segments of a body with anatomically-meaningful boundaries, like upper arm, anterior thigh, etc. [12] or by selecting regions based on known referral patterns on dermatomal/sclerotomal/peripheral nerve maps [34].

- **Shading:** Participants identify their pain by painting or scribbling to shade their pain locations [35].

- **Checklist:** Pain mapping carried out by check-listing predefined body locations to indicate pain experience, which might include a checkbox [36].

## Modes of implementation of pain/symptom maps:

- **Pen-on-paper:** Pain maps are delivered manually as a 2D/3D sketch/image on plain paper for the respondent to indicate their pain location or extent [37].

- **Real Manikin:** Pain mapping carried out on a three-dimensional human-like (real) dummy manikin with physiological structure [38].

- **Digital:** Respondents assessed a pain map as a scanned image or a computer-generated image through mobile apps, tablets using stylus or touch screens [37] including personal computer devices using mouse click [31].

**Pain measure.** These are the pain characteristics being measured by the pain/symptom mapping technique and could be a measure of pain extent or area, pain location or distribution (spatial); pain intensity, severity or quality (nociception)

### Critical appraisal of individual sources of evidence

The review team will present a narrative synthesis of evidence and does not intend to conduct quality appraisal for included studies due to the broadness of the review question, and wanting a broad scope of study designs to be included in the review, which is intended to completely examine existing evidence, identify literature gaps and provide a comprehensive scope as a preliminary step for a systematic review. This aligns with Peters and colleagues [26], who posited that it is not usually necessary to undertake a methodological appraisal of the sources included in a scoping review because it seeks to develop a comprehensive overview of the evidence rather than a quantitative or qualitative synthesis of data.

### Synthesis of results

The most commonly used types, methods, and modes of assessing musculoskeletal pain characteristics with pain/symptom maps will be analyzed through categorization by common features and reported through descriptive statistics (e.g., mean number of body regions, proportion of all literature using a particular mode of implementation). Types will be synthesized by exploring the scale, body region, $n$-dimensional orientation, segmentation, and sex identifications of the implemented pain/symptom mapping tools explored in the included articles.

Methods will be analyzed by assessing how pain characteristics are measured, reported, and evaluated in the included studies, e.g., marking, grid, shading, and checklist. Modes will be examined as to how the musculoskeletal pain participants interact with the pain/symptom mapping tools, in the included studies, e.g., pen-on-paper, digital, dummy manikin (real human-like model), or a combination of processes. Types of pain metrics extractable (e.g., extent, location, intensity, quality) will be reported in summary tables. The synthesis of the results will rely on both descriptive statistics and narrative synthesis. Descriptive statistics will summarize quantitative data, such as the frequency and distribution of different types, methods, and modes of pain mapping. Narrative synthesis will be used to integrate and describe qualitative findings, focusing on trends, patterns, and variations across the studies. This approach will allow for a comprehensive understanding of the literature, identifying key insights as well as inconsistencies or gaps in the research.

### Patient partner engagement

The development of the protocol has been discussed with a Patient Partner Advisory Group (PPAG) associated with the CANSpine research group at Western University (London Ontario, Canada). They will be invited to contribute feedback and suggestions periodically throughout the conduct of the review and the interpretation of the results. This protocol has been presented to the PPAG and the group indicated broad support for the importance and impact of the project, while suggesting minor opportunities for improvement (e.g., suggesting areas for deeper investigation or focus). The PPAG will continue to receive periodic updates on the project, after the initial round of interpretation and again prior to submission of the final manuscript. Their feedback will be critical for shaping the focus and messaging from this review.

## Discussion

Pain maps have historically been viewed as adjunctive tools for clinicians and researchers to better understand a patient's pain through largely qualitative means and have accordingly often been designed without rigorous quantification in mind. Hägg and colleagues [39] found that the predictive value of pain drawing in predicting outcomes for surgical and non-surgical interventions is not properly documented. Conversely, some recent studies have shown that computational quantification of pain maps has the potential to enhance comprehensive pain assessment, facilitating research and personalized care over time for patients with various musculoskeletal pain conditions [40,41]. Similarly, pain mapping has significantly improved pain reporting [42], and tracking the effectiveness of treatment pathways [23].

Applying the methodology of a scoping review in collating historical and current ideas in the type, methods, and modes of implementing pain maps will potentially mitigate the inconsistencies observed in musculoskeletal pain drawing. However, in keeping with the broad scope of this review, there is a limitation in language inclusivity. This review focuses on identifying the characteristic features of common pain maps and scoring strategies, which are typically reported in multiple publications, although the highest-impact journals in the field require, at a minimum, an English-language translation. Additionally, included studies will not be typically assessed for quality as there are no standards for operationalizing a pain map which is probably deployed differently in different study designs [42].

As pain mapping evolves from a qualitative tool to a more quantifiable and data-driven resource, it holds considerable promise in both personalized patient care and research outcomes. In clinical settings, pain maps can enhance personalized patient care as a novel communication channel between patient and clinician, as a means of quantifying spatial pain

measures to optimize treatment targeting, and enabling clinicians to track progress and evaluate treatment effectiveness beyond the historic reliance on only pain intensity ratings. A more structured approach to pain mapping can complement patient narratives, fostering consistent, patient-centric and evidence-based recommendations. The potential for personalized care is amplified by advancements in artificial intelligence (AI) and machine learning (ML), which when combined with standardized pain maps can detect subtle pain patterns, predict outcomes, and suggest personalized treatment plans [17]. Integrating pain maps into electronic health records (EHRs) enhances the holistic view of a patient's health [18]. From a research perspective, the scoping review will synthesize existing pain mapping methodologies and identify gaps, guiding future studies. Standardizing pain map protocols across studies would enhance comparisons and collaboration, advancing pain management research.

This scoping review is intended to provide a clear picture of the prior use and current state of pain/symptom maps in the musculoskeletal pain field. As technology is rapidly evolving and core data sets are becoming increasingly important for making decisions at the individual and population health level, there is a clear window of opportunity for the development, implementation, and interpretation of pain/symptom maps to also evolve in new ways that contribute unique and relevant data across health conditions. Notably, the rise of increasingly accessible artificial intelligence and machine learning platforms appears to signal that the information gleaned from pain/symptom maps could go much further than a simple summation of pixels or body regions. As a first step towards this level of deep rigor and an 'ideal' pain/symptom map, we find it prudent to first present the scope and summarize the historic and recent trends that have led the field to where it is today.

## Future direction

This review will describe and synthesize recent and current trends, uses, modalities, and gaps in the use of pain mapping for research or clinical use. This is a critical step towards the development of more meaningful and informative maps that leverage new and emerging technologies while centering patient values and needs related to communicating pain. Next steps are expected to include presentation of results to the PPAG, linkages with experts in the field of machine learning, and rigorous design and testing of novel approaches to pain mapping.

## Supporting information

**S1 File. PRISMA-P (Preferred Reporting Items for Systematic review and Meta-Analysis Protocols) 2015 checklist: recommended items to address in a systematic review protocol\*.** (DOCX)

**S2 File. Search strategy framework (Medline).** (DOCX)

**S3 File. Data extraction form.** (DOCX)

## Author contributions

**Conceptualization:** Ukponaye Desmond Eboigbe, David M. Walton.

**Data curation:** Ukponaye Desmond Eboigbe, Aliyu Lawan.

**Formal analysis:** Ukponaye Desmond Eboigbe.

**Investigation:** Alison Rushton, David M. Walton.

**Methodology:** Ukponaye Desmond Eboigbe.

**Project administration:** Alison Rushton, David M. Walton.

**Supervision:** Alison Rushton, David M. Walton.

**Validation:** Aliyu Lawan, David M. Walton.

**Writing – original draft:** Ukponaye Desmond Eboigbe.

**Writing – review & editing:** Aliyu Lawan, Alison Rushton, David M. Walton.

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
