## [Decision Letter · Decision Letter 0]

19 Dec 2024

PONE-D-24-32938Types, Method, and Mode of Implementation of Pain/Symptom Maps in Musculoskeletal Pain Rehabilitation: A Scoping Review ProtocolPLOS ONE

Dear Dr. Eboigbe,

Thank you for submitting your manuscript to PLOS ONE. After careful consideration, we feel that it has merit but does not fully meet PLOS ONE’s publication criteria as it currently stands. Therefore, we invite you to submit a revised version of the manuscript that addresses the points raised during the review process.

We look forward to receiving your revised manuscript.

Kind regards,

Nik Hisamuddin Nik Ab. Rahman

Academic Editor

PLOS ONE

**Journal Requirements:**

Reviewers' comments:

Reviewer's Responses to Questions

**Comments to the Author**

1. Does the manuscript provide a valid rationale for the proposed study, with clearly identified and justified research questions?

Reviewer #1: Yes

Reviewer #2: Yes

2. Is the protocol technically sound and planned in a manner that will lead to a meaningful outcome and allow testing the stated hypotheses?

Reviewer #1: Partly

Reviewer #2: Yes

3. Is the methodology feasible and described in sufficient detail to allow the work to be replicable?

Reviewer #1: Yes

Reviewer #2: Yes

4. Have the authors described where all data underlying the findings will be made available when the study is complete?

Reviewer #1: Yes

Reviewer #2: Yes

5. Is the manuscript presented in an intelligible fashion and written in standard English?

Reviewer #1: Yes

Reviewer #2: Yes

6. Review Comments to the Author

You may also provide optional suggestions and comments to authors that they might find helpful in planning their study.

**Reviewer #1: ** Suggestions for Improvement

Expand Inclusion Criteria:

Consider including studies in other major languages, utilizing translation tools or services to mitigate resource constraints, thereby enriching the diversity of data.

Clarify Patient Involvement:

Provide more details about how the Patient Partner Advisory Group (PPAG) will be engaged throughout the review process, beyond feedback and suggestions.

Address Applicability:

Expand the discussion on how the findings will be practically applied in clinical and research settings, emphasizing their potential impact on personalized patient care.

**Reviewer #2:**  Abstract

Objective: The abstract should clearly specify the review timeframe, key databases, and unique contributions compared to prior studies.

Outcome Clarity: Add distinct primary and secondary outcomes.

Introduction

Please clearly define the distinction between "methods" and "modes" of implementation to avoid reader confusion.

Highlight the role of advanced technologies, such as machine learning, more explicitly.

Methods

Please Clarify exclusion criteria for artistic body maps and imaging modalities. Examples would be helpful.

Consider justifying why non-English studies are excluded rather than providing translations or summaries, as this may limit generalizability.

Search Strategy:

Include an explicit description of the keywords and Boolean operators used in the search strategy.

The framework of the search strategy could benefit from more examples of "grey literature" sources.

Screening and Selection:

Please add more detail on the process of resolving conflicts between reviewers during screening.

Specify whether quality checks (e.g., inter-rater reliability) will be conducted on reviewer decisions.

Data Extraction:

Expand on how the data extraction form will evolve during the pilot phase, detailing examples of proposed modifications.

Results and Synthesis

Clearly state whether thematic analysis will be incorporated into the synthesis or if results will rely purely on descriptive statistics. The manuscript mentions narrative synthesis but does not outline a clear strategy for integrating findings across diverse study designs.

Discussion

The authors rightly discuss the limitations of resource constraints. However, elaborating on strategies to mitigate these, such as collaboration with translators, would be beneficial.

Future Directions:

Provide concrete recommendations for future research based on the potential findings.

References

The references are comprehensive but verify that they adhere to the journal’s citation format, particularly for online resources.

7. PLOS authors have the option to publish the peer review history of their article (what does this mean? ). If published, this will include your full peer review and any attached files.

**Do you want your identity to be public for this peer review?** For information about this choice, including consent withdrawal, please see our Privacy Policy .

Reviewer #1: No

Reviewer #2: No

---

## [Author Response · Author response to Decision Letter 0]

30 Jan 2025

A separate file (titled: Response to Reviewers) addressing the reviewers' comment had been attached.

Thank you.

---

## [Editor Report · Decision Letter 1]

4 Feb 2025

Types, Method, and Mode of Implementation of Pain/Symptom Maps in Musculoskeletal Pain Rehabilitation: A Scoping Review Protocol

PONE-D-24-32938R1

Dear Dr. Eboigbe,

We’re pleased to inform you that your manuscript has been judged scientifically suitable for publication and will be formally accepted for publication once it meets all outstanding technical requirements.

Kind regards,

Nik Hisamuddin Nik Ab. Rahman

Academic Editor

PLOS ONE
---

## [Editor Report · Acceptance letter]

PONE-D-24-32938R1

PLOS ONE

Dear Dr. Eboigbe,

I'm pleased to inform you that your manuscript has been deemed suitable for publication in PLOS ONE. Congratulations! Your manuscript is now being handed over to our production team.

Kind regards,

on behalf of

Professor Dr Nik Hisamuddin Nik Ab. Rahman

Academic Editor

PLOS ONE